

# Potential of INSAT-3D Sounder Derived Total Precipitable Water Product for Weather Forecast

Shailesh Parihar[*], Ashim Kumar Mitra and [a]Rajiv Bhatla

*India Meteorological Department, New Delhi-110003*

*[a]Banaras Hindu University, Varanasi-221005*

*Email- shellsalpha@gmail.com

## Abstract

INSAT-3D satellite objectives to upgrade the meteorological observation, monitoring of earth surface atmosphere for weather forecasting and disaster warning. The amount of the water vapor present in atmospheric column in the form of total precipitable water (TPW) derived product from atmospheric sounding system is one such weather monitoring capability in the INSAT-3D payload. The current study is based on INSAT-3D satellite sounder derived TPW and corresponding TPW from radiosonde observations (RS) and National Oceanic and Atmospheric Administration (NOAA), N-18 and N-19 have been used to assess retrieval performances. The RS TPW from 34 India Meteorological Department (IMD) stations over the Indian region from May to September 2016 has been considered for the validation. The analysis is performed on daily, monthly, sub-divisional and overall basis over the Indian region. On daily and monthly scale against RS TPW, the root mean square error (RMSE) and correlation coefficients (CC) of INSAT-3D TPW are in and around of 8 mm and above 0.8 respectively. However, on sub-divisional and overall scale, the RMSE found to be in the range of 1 to 2 mm and CC was around 0.9 in comparison with RS and NOAA. The spatial distribution of INSAT-3D TPW with actual rainfall observation is also been investigated. In general, INSAT-3D TPW correspond well with rainfall observation however, heavy rainfall events occurs in the presence of high TPW values. Furthermore, a case study with INSAT-3D TPW and ground based Global Navigation Satellite System (GNSS) receiver network have been demonstrated. It is noticed that, INSAT-3D TPW can be considered as a precursor for mesoscale activity very well.

The purpose of this study is to investigate the potential use of operational INSAT-3D sounder derived TPW to weather forecast. However, the major source of improvement in INSAT-3D TPW is mainly applying the GSICS calibration corrections (Global Space-based Inter-Calibration



System) on Infra-Red (IR) sounder channels at IMDPS, New Delhi, which aims to produce
corrections, ensuring the data consistency and allowing them to be used to produce globally
homogeneous products for environmental monitoring. The current TPW from INSAT-3D satellite
can be utilized operationally for weather purpose and it can also offer substantial opportunities for
improvement in now casting studies.
**Keywords:** INSAT-3D Sounder, Total Precipitable Water, rain fall.

# 1. INTRODUCTION

Water vapour is one of the most variable characteristics of the atmosphere. It is an essential factor
in climate and weather. It regulates air temperature by absorbing thermal radiation both from the
Sun and the Earth; it is directly proportional to the latent energy available for the generation of
storms; and it is the ultimate source of all forms of condensation and precipitation. Total
Precipitable Water (TPW) Vapor data are potentially significant for weather and climate modeling
and prediction. Kuo et al., (1996) found significant improvement in precipitation forecasts when
TPW data are assimilated in numerical weather prediction models. Yuan et al., (1993) simulated
the utilization of TPW data in monitoring global and regional climate change and found up to 8
mm increment in tropical TPW resulting from doubling of atmospheric $CO_2$.
TPW which may be monitored from meso-scale to large scale convective activity and its relation
to synoptic circulation systems such as Thunderstorm, monsoonal activities, asymmetries and
moisture gradients that aid in interpreting tropical cyclones interaction with dry/moist air by high
convective cloud bases. It is the source of the latent heat which is released into the atmosphere
during cloud formation. It also dominates the structure of diabatic heating of the earth's
atmosphere (Trenberth et al., 2005; Trenberth and Stepaniak, 2003a, b). These parameters vary in
time and as well as in space (both vertically and horizontally) throughout the atmosphere. In global
atmospheric analyses, the problems come with spatial and temporal gaps. They were impossible
to use for climate purposes owing to the huge discontinuities in time as the inevitable
improvements were incorporated into the analyses (Trenberth and Olson 1988). This could be
possible with accurate temperature and moisture profile either from in-situ observations or
remotely sensed data for weather forecasting and nowcasting purposes.



The INSAT-3D satellite imager and sounder products major play role in weather monitoring and provides new understanding of the atmospheric process for monitoring weather. Especially vertical profiles of temperature, humidity and ozone content in atmosphere. It improves analytical skill for weather forecasters. However, the INSAT-3D sounder data is being derived for generally clear sky conditions (Ratnam et. al., 2016). Its wide variety of products such as Geopotential height which can be used to analyze the structure of the convective system TPW, can provide details about mesoscale phenomena and tropical cyclone structure, eg. Symmetries and moisture gradients that aid in interpreting tropical cyclones interaction with dry/moist air.

In this study, the INSAT-3D sounder TPW product was statistically compared with radiosonde observations and NOAA satellite data over the period May–September 2016. The purpose of this comparison is to investigate the potential use of operational hourly derived INSAT-3D sounder TPW for monitoring of particularly on severe weather phenomenon over the Indian region. However, initial work using INSAT-3D sounder data has been done by Mitra et al. 2015, who compared INSAT-3D data obtained at 10 RS stations of IMD and Ratnam et al. 2016, using 34 RS observations and compared with data from other satellites like AIRS, MLS and SAPHIR and from ERA-Interim and NCEP re-analysis data sets. Which concludes that INSAT-3D provides measurements with very good spatial and temporal coverage over India compared to any other satellites. It is found that there is a large difference between INSAT-3D and other data sets both in temperature and water vapour above 25˚N latitude perhaps due to their geometry. In the present paper, we extended the work with 34 RS stations and taking NOAA data on daily, monthly, sub divisional scale followed by a case study with IMD installed network of GNSS TPW. Furthermore, the spatial distribution of INSAT-3D TPW with actual rainfall observation has also been investigated.

## 2. DATA BASE

### 2.1 INSAT-3D Sounder Scan strategy in IMD

*2.1.1 INSAT-3D Sounder Specification*

INSAT-3D sounder is advance weather satellite with improved imaging system and atmospheric sounder. INSAT-3D sounder generating vertical profile of the atmosphere in terms of temperature



and humidity. INSAT-3D payloads such as sounder is first time introduce in ISRO satellite
mission. INSAT3D sounder has one visible spectral channel and eighteen channels in shortwave
infrared (SWIR), middle infrared (MIR) and long wave infrared (LIR) regions. For all channels
ground resolution is 10x10km. The INSAT-3D sounder detail can be found on Mitra et.al, 2015.

92                                  **Table 1Sounder Specification**

| Channels (Spectral Range Microns) | Resolution |
|---|---|
| Visible (0.67) | 10X10 Km |
| SWIR (3.67) | 10X10 Km |
| MIR (6.38) | 10X10 Km |
| LWIR (11.66) | 10X10 Km |

*2.1.2 INSAT-3D Sounder Scan Strategy*
INSAT-3D retrieval algorithm at IMDPS, New Delhi is designed for retrieving vertical profiles of
temperature and moisture from radiances in 18 channels. National Center for Environment
Prediction (NCEP) Global Forecast System (GFS) consisting of 12 hour forecast, retrieval
algorithm has taken for first-guess input. On account of its higher temporal as well as horizontal
resolution, these retrievals provide the significant data on vertical structure of the atmosphere over
land and oceanic region. Currently, these information are being obtained from conventional in-situ
based methods of observations. INSAT-3D retrieval algorithm is based on clear sky infrared
radiances, which includes different absorption bands observed through sounder all channels on an
hourly time scale. In the scheme, computation of the hybrid first guess atmospheric profiles using
a linear combination of regression retrieval and NWP model forecast are being used (Mitra et al.,
2015). The methodology has followed by non-linear physical retrieval procedure (Li et al., 2000;
Ma et al., 1999) for first guess consistent with the sounder observations. The another methodology,
pressure layer fast algorithm for atmospheric transmittance (PFAAST) radiative transfer model
(Hanon et al., 1996) has used for the forward computation of sounder channel radiances along with
the Jacobians, which is used in the physical retrieval.
Figure-1 shows that every hour over Indian land mass (A) and every 6th hour Southern Hemisphere
(B), the sounder data is being processed at IMDPS, New Delhi on an operational basis. This is the





simplest scanning mode kept in such a way that sounding over larger region (land+ocean) will be
available every hour. Full frame mode scan is 18º × 18º North South (NS) covering the entire earth
disc in about 25.7 minutes. Program mode - 18º in East West (EW) direction, while NS coverage
can be programmed. Sounder completes sounding in 10 km × 10 km area in 0.1s and performs
space look operation once every 2 minutes. Black body calibration is performed in every 20
minutes or on command. INSAT-3D Sounder have a capability to scan in the steps of 64 × 64
pixels. Scanning of a region covering 640 × 640 pixels that is roughly 6400 km × 6400 km will
take about 180 minutes. This implies that the one step of 64 × 64 will take approximately 1.8
minutes. The INSAT-3D provides hourly observations over Indian landmass and every 6-hour
observations over Oceanic region. The uncertainties can occur in quantitative accuracy in the
precipitation estimation during nonsummer seasons, where convective precipitation is not
dominant (Arkin and Xie 1994). This benefits of these kind of scan strategy is best utilize for the
studies of initial convections and genesis of evolution of squall lines and there fine structure
(Purdom 1996 a). Therefore, INSAT-3D sounder scan strategy can be used for nowcasting and
NWP model assimilation over Indian region.



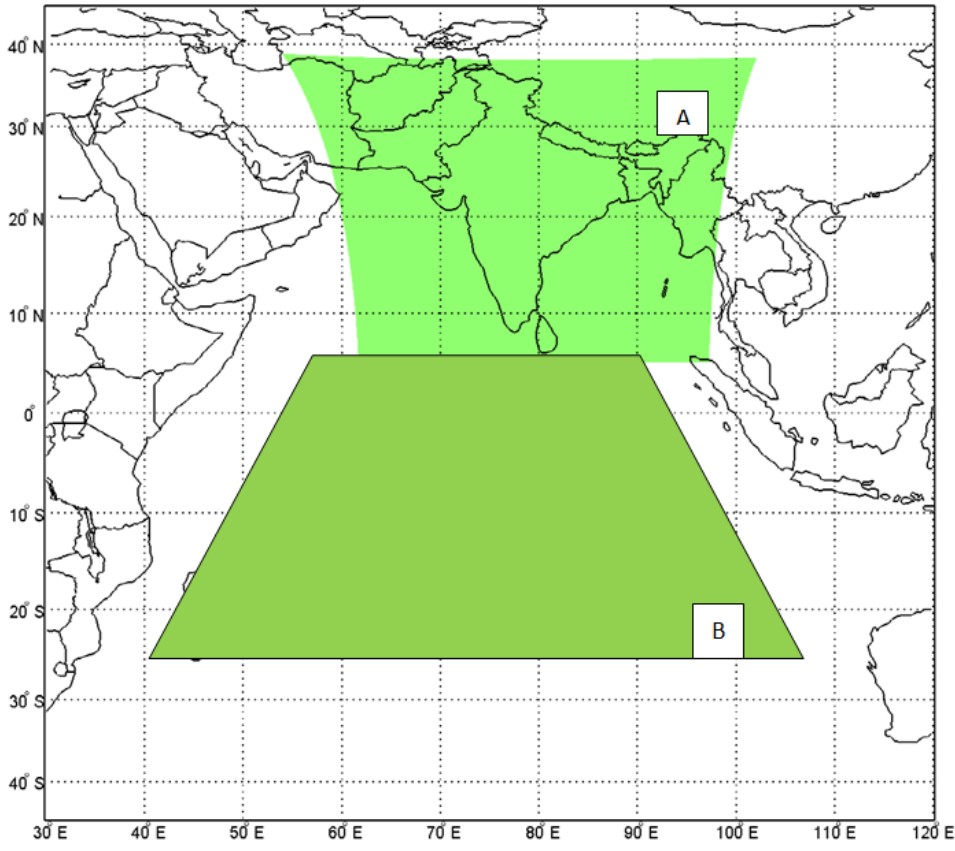


**Figure 1. INSAT-3D Sounder scan processing strategy over land and ocean.**

**2.2 Radiosonde Observations (RS)**
In IMD, upper air observations are made at 43 RS/RW stations, 34 RS/RW stations are being used
and 62 Pilot Balloon observatories to provide pressure, temperature, humidity & wind at various
levels in the atmosphere up to an altitude of 30-35 kms for RS/RW and up to a maximum altitude
of 18 kms employing optical theodolites for PB stations. The types of ground equipment used in
RS observatories have been equipped with three types of ground equipment as under;-
• Radiosonde Ground equipment (ECIL/DIGITAL make) along with X band Win
• d finding Radars (EEC/MULTIMET) at 401 MHz
• IMS-1500 Radiotheodolite at 1680 MHz
• SAMEER Radiotheodolite at 401 MHz





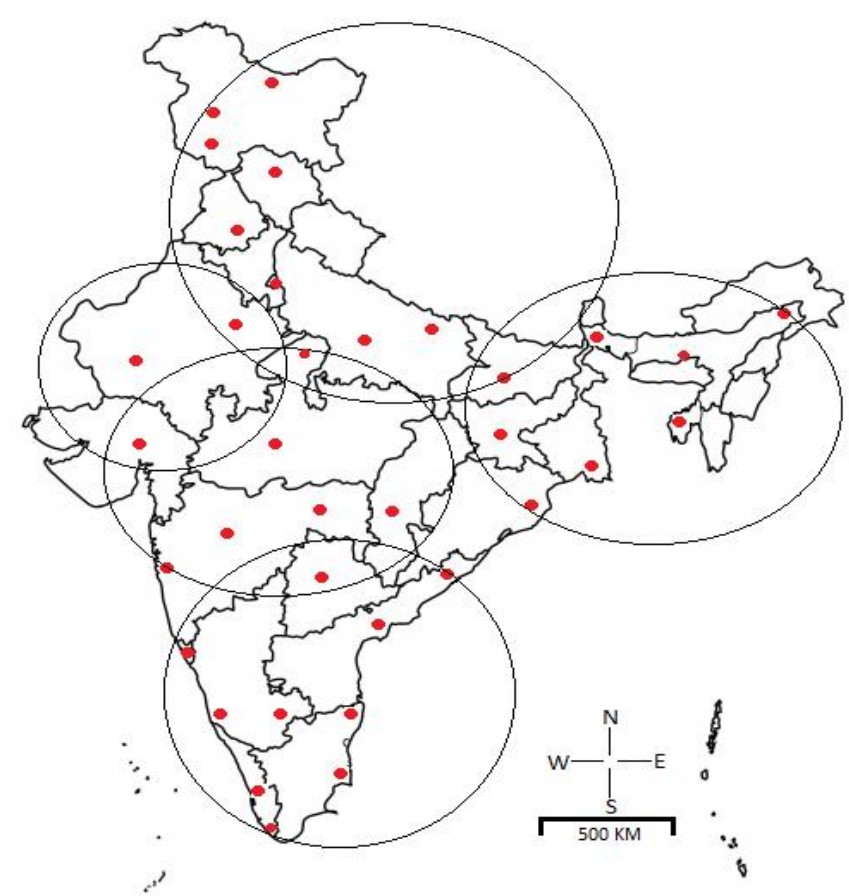


**Figure 2. Radiosonde Stations of IMD over India**

Over all 34 RS stations marked in red color in Figure-2 is taken for the comparison with INSAT-
3D TPW.
**2.3 NOAA**
NOAA's National Environmental Satellite Data and Information Service works for weather
community launched Advanced Microwave Sounding Unit (AMSU) was aboard in the National
Oceanic and Atmospheric Administration's (NOAA) polar–orbiting satellites. The first AMSU–A
was launched on the NOAA–15 satellite on 13 May 1998. NOAA real time data have been
downloaded from www.nnvl.noaa.gov.in.
**2.4 GSICS based inter-calibration**



There is an on-board blackbody which is responsible for generation of calibration information
for all the IR channels in Imager and Sounder payloads. In-orbit readings of blackbody precision
temperatures revealed a gradient among the sensor readings which led to inaccuracy in getting
the correct blackbody temperature. It was also observed that during satellite midnight, sun-rays
from behind the earth enter directly into the sensor and hence lead to increase in blackbody
temperatures. This phenomenon also leads to incorrect calibration information generation. In order
to provide climate quality products and to improve the calibration information quality
GSICS        based        inter-calibration        is        used        for        INSAT-3D/3DR.
Thus the channel wise GSICS coefficient found out are absorbed in the Radiometric Correction
process. These coefficients are applied on the channel-wise radiances found out by
using in-orbit calibration information in the retrieval process.

## 3. METHODOLOGY

INSAT-3D retrieval algorithm at IMDPS, New Delhi, is designed for retrieving vertical profiles
of atmospheric temperature and moisture in the atmosphere from clear sky infrared radiances in
different absorption bands observed through the various sounder channels on an hourly time scale.
India Meteorological Department (IMD), New Delhi, has adapted sounder retrieval scheme from
the operational High resolution Infrared Radiation Sounder (HIRS) processing scheme and
Geostationary Operational Environmental Satellites (GOES) algorithms developed by Cooperative
Institute for Meteorological Satellite Studies (CIMSS), University of Wisconsin, USA (Ma et al.,
1999 and Li et al., 2000). In this system, physical and regression based retrievals are employed.
Which includes spectral bands in and around the $CO_2$ and $H_2O$ absorbing bands. These bands
having the vertical structure of atmospheric temperature and moisture (Mitra et al. 2015).
The Sounder-derived product like Total Precipitable Water (TPW) is the amount of the water vapor
present in atmospheric column from the surface of earth to space.  Mathematically, if a(p) is the
mixing ratio at the pressure level, p, then the INSAT-3D precipitable water vapor  W, contained
in a layer bounded by pressures $p_1$ and $p_2$ is given by
$$\text{INSAT3D Precipitable Water Vapor} = \frac{1}{\rho g} \int_{p1}^{p2} a \, dp$$



Where ρ represents the density of water and g is the acceleration of gravity and can be found in
detail http://www.imd.gov.in/INSAT-3D/categouge.
In the present study, we are being done RS observations for pairing with their closest INSAT-3D
TPW values. The RS observations provide temperature and dew point as functions of pressure at
both mandatory and significant levels and these data were obtained from IMD's network. The
performance of IMD's GPS radiosonde stations has been very well examined using ECMWF
global data monitoring report by Gajendra Kumar et al., (2011).
In our evolution methodology, each RS was paired with closest INSAT-3D TPW and patterned
according to criteria suggested in Fuelberg and Olson (1991). The collocation criteria for INSAT-
3D retrievals with RS and NOAA data are based on the following: (1) The absolute distance
between the position (latitude and longitude) of the RS and the INSAT-3D retrievals has been
considered as 0.5º (50 Km). This will minimize the differences arising from horizontal gradients.
(2) The temporal difference between two sets of data is around ±120 minutes depending on
retrievals and location of the RS station.
Furthermore, to improve the quality of sounder derived radiances in INSAT-3D, the Global Space-
based Inter-Calibration System (GSICS) corrections have been applied to the INSAT-3D sounder
radiances on an operational basis at IMDPS, New Delhi. The GSICS aims to inter-calibrate a
diverse range of satellite instruments to produce corrections ensuring their data are consistent,
allowing them to be used to produce globally homogeneous products for environmental
monitoring. GSICS develops common methodologies to check the quality of sensors operated by
various satellite agencies over the worldwide. These satellite organizations works for weather
forecasting, climate & atmosphere changing and environmental applications. This is post launch
calibration strategy which involves spectral response function of sensors, sensor performances and
inter-calibration of satellite sensor after that recalibration of archived data or products of sensors.

202              **3.  RESULTS AND DISCUSSIONS**




## 4.1 Comparison of INSAT-3D with RS and NOAA TPW at Daily, Monthly and Sub divisional Scale

INSAT-3D derived TPW is being derived over Indian region almost every hour. For verification purposes of NWP model derived precipitable water and usefulness in weather forecast. It is desirable to compare INSAT-3D TPW at different time scale with different sets of data. Thus, on a daily scale, we compared the INSAT-3D TPW with all the collocated datasets against RS TPW. On monthly scale, month average data on collocated points were averaged. For sub-division scale, five different regions categorized according to meteorological subdivisions are, Northern India (NI), Eastern India (EI), Central India (CI), Western India (WI) and Peninsular India (PS) (figure 2).

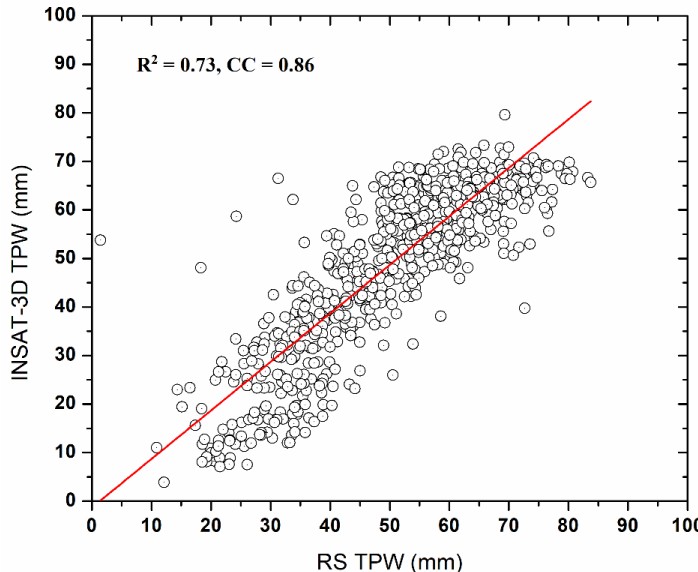

**Figure 3. INSAT-3D sounder TPW with RS for Day-wise from May to September 2016**

Figure 3 shows the comparison of INSAT TPW and RS TPW on daily scale during May-September 2016. On a day to day basis, INSAT-3D TPW agrees well with RS TPW. There is a consistent correlations noticed above 40mm of TPW values whereas less than 40mm TPW values, INSAT-3D underestimates the RS observations. This may be attributed to seasonal variation, orographic of the region and different climatic zone over India. The overall correlation on daily scale was found to be 0.86. In the previous study, Mitra et al. (2015) have reported 0.73 correlations



using 10 IMD stations. Figure 4 shows the comparison of INSAT-3D TPW and RS TPW on
Monthly scale during May-September 2016. It can be noticed that during monsoon period and
specially month of June, July and August, heavy rainfall (above 64.5 mm) occurrence have very
well agreement with INSAT-3D TPW values. Mostly INSAT-3D TPW is higher when rainfall
occurrence is higher above 40 mm. Monthly correlation was found to be above 0.78.
In table 2, INSAT-3D and RS Relation between the standard deviation and the arithmetic mean
values indicates that the deviation from the normal distribution cannot be ignored. INSAT-3D
coefficient of variation values are high as compared with RS, which indicates that total precipitable
water being variable measured by INSAT-3D. The lower the coefficient of variation of TPW
amount in July to September, 2017. The Coefficient of skewness values found negative between
INSAT-3D and RS measurement, which indicates mean is less than the mode of the data. The
correlation coefficient show good agreement with well RMSE for June to September, 2017 except
in month of July. The student's t-test has calculated for significance of computed parameter. Which
is provided distribution of same is normal for INSAT-3D and RS measured data. This found to
good fit between, INSAT-3D and RS TPW.

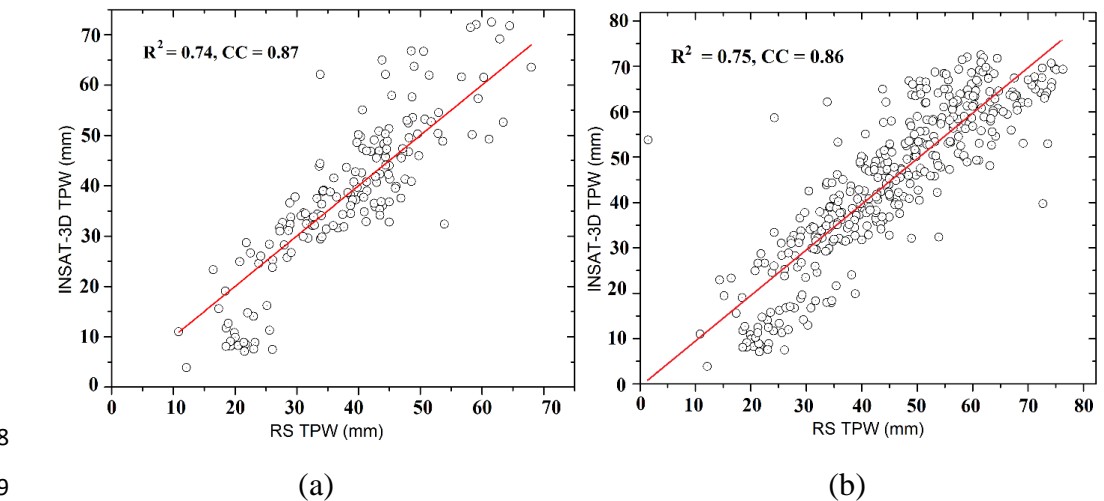


239                         (a)                                             (b)





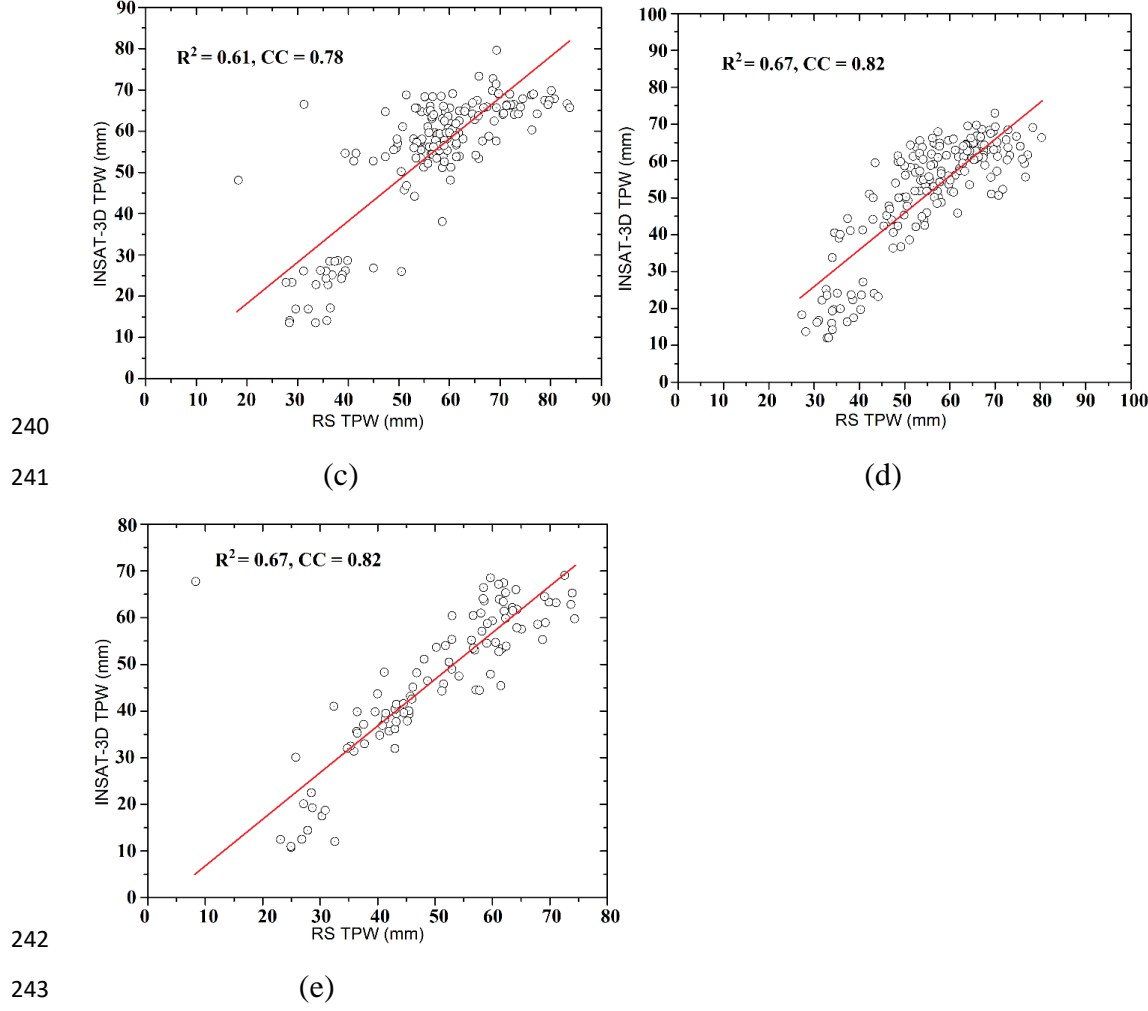


241                             (c)                                                          (d)


243                             (e)

**Figure 4. INSAT-3D sounder TPW with RS for (a) May (b) June (c) July (d) August and (e)**
**September 2016**

**Table 2. Statistics and correlation between total precipitable water measured by INSAT-3D**
**and RS**

| Month | INSAT-3D | RS | INSAT-3D | RS | INSAT-3D | RS | INSAT-3D | RS | CC | RMSE (mm) | t-test |
|---|---|---|---|---|---|---|---|---|---|---|---|
| | Arithmetic Mean (mm) | | Standard Deviation | | Coefficient of Variation | | Coefficient of Skewness | | | | |
| **May** | 39.36 | 39.87 | 15.40 | 12.51 | 0.39 | 0.31 | -0.21 | -0.10 | 0.87 | 7.69 | 0.359931 |



| | | | | | | | | | | |
|---|---|---|---|---|---|---|---|---|---|---|
| **Jun** | 49.75 | 52.66 | 16.44 | 14.16 | 0.33 | 0.26 | -0.87 | -0.57 | 0.86 | 8.50 | 0.049282 |
| **July** | 54.87 | 60.44 | 14.59 | 12.53 | 0.26 | 0.20 | -1.45 | -0.61 | 0.78 | 9.31 | 0.000012 |
| **Aug** | 52.09 | 57.33 | 14.71 | 11.97 | 0.28 | 0.20 | -1.24 | -0.49 | 0.82 | 8.73 | 0.000022 |
| **Sep** | 49.00 | 54.30 | 14.14 | 13.69 | 0.28 | 0.25 | -1.01 | -0.31 | 0.82 | 8.79 | 0.000213 |


Figure 5 shows the comparison of INSAT-3D TPW with RS and NOAA TPW on sub divisional
scale during May-September 2016. It can be clearly seen from the figures that INSAT-3D TPW is
under estimating the RS TPW whereas it is over estimating the NOAA TPW for the entire region
during the monsoon period. However, the moisture convergence, advection of moisture and
geographical locations of the subdivisions, which occasionally receive heavy to very heavy rainfall
due to synoptic scale monsoon circulations or orographic area during summer monsoon season
plays the major role in the activity of rainfall over the Indian sub-continent. The areas such as high
orographic region like north eastern parts and Jammu and Kashmir and parts of Western Ghats in
the west coast of India, have less evaporation rates and high rainfall and the moisture laden that is
transported into the region could not be fully captured by the satellite observations. Similarly it is
also observed that the rainfall is overestimated in the dry conditions because the falling raindrop
evaporates before coming to the surface in dry conditions resulting the over estimation while
comparing with actual observation. In contrast, CI and PS, have a good correlations compared to
EI and NI. However, opposite trend were found while comparing INSAT-3D TPW with NOAA
TPW. INSAT-3D TPW is always higher over NOAA data. One of the reasons is that INSAT-3D
sounder derived products were calculated from the radiances sampled every hour while NOAA
TPW were based on only two satellite passes with equator crossing times of 0230 and 1430 local
time. Therefore, TPW derivation methodology is the same whether the radiometer is on polar or
geostationary satellite, the sampling frequency of the radiometer is much higher in a geostationary
satellite than polar satellite. In general, sub-divisional comparison revels that the INSAT-3D TPW
agrees well RS and NOAA TPW below 23°N whereas the difference is higher above 23°N.
In table-3. The standard deviation and the arithmetic mean values indicates that the deviation from
the normal distribution, cannot be ignored, which shown deviation according from the data with
subdivisions in India for INSAT-3D, RS and NOAA. INSAT-3D coefficient of variation (CV)
values are similar with RS, which indicates that total precipitable water being variable, but in case




of NOAA CV found higher with respect to INSAT-3D and RS. The Coefficient of skewness values
found negative in case INSAT-3D, RS and NOAA measurement, which indicates mean is less than
the mode of the data. The correlation coefficient (CC) show good agreement with well RMSE
from June to September, 2017 between INSAT-3D and NOAA found 0.96 and INSAT-3D and RS
found 0.87.

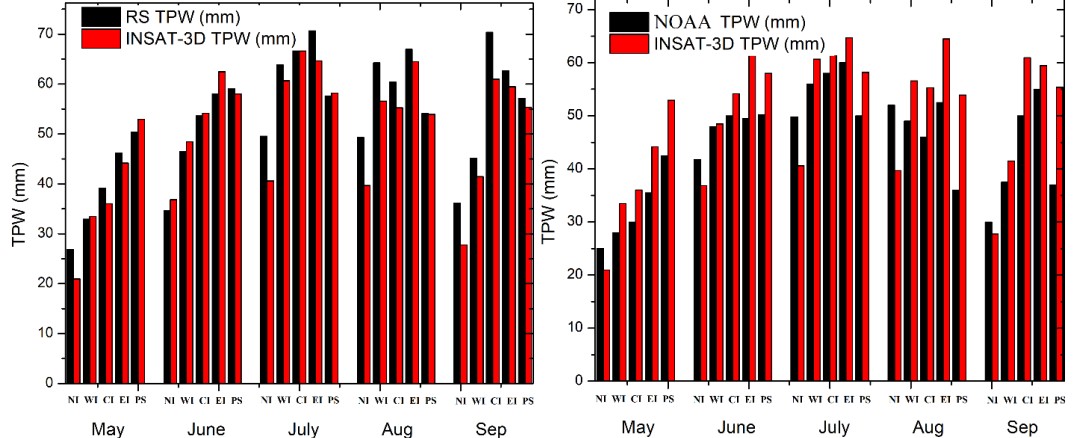


**Figure 5. Subdivision wise NI, WI, CI, WI & PS from May to September 2016 between**
**INSAT-3D and RS (left), NOAA (right)**
**Table 3. Statistics for total precipitable water measured by INSAT-3D, RS and NOAA**
**sub divisional regions of India**

| Sub div. | Sensors | Arithmetic Mean | SD | Coefficient of Variation | Coefficient of Skewness | NOAA vs INSAT-3D | | | INSAT-3D vs RS | | |
|---|---|---|---|---|---|---|---|---|---|---|---|
| | | | | | | BIAS | RMSE | CC | BIAS | RMSE | CC |
| NI | NOAA | 39.71 | 11.91 | 0.30 | -0.29 | | | | | | |
| | INSAT-3D | 33.16 | 8.51 | 0.25 | -0.84 | 1.3 | 1.09 | 0.97 | 1.22 | 1.15 | 0.87 |
| | RS | 39.28 | 9.91 | 0.25 | 0.005 | | | | | | |
| WI | NOAA | 43.7 | 10.98 | 0.25 | -0.63 | | | | | | |
| | INSAT-3D | 48.13 | 11.04 | 0.22 | -0.26 | -0.88 | 0.88 | 0.97 | 0.47 | 0.77 | 0.97 |
| | RS | 50.52 | 13.42 | 0.26 | -0.12 | | | | | | |
| CI | NOAA | 46.8 | 10.35 | 0.22 | -1.22 | | | | | | |
| | INSAT-3D | 54.61 | 11.51 | 0.21 | -1.20 | -1.56 | 1.23 | 0.97 | 0.79 | 0.83 | 0.96 |
| | RS | 58.58 | 12.83 | 0.21 | -0.90 | | | | | | |





| EI | NOAA | 50.5 | 9.22 | 0.18 | -1.28 | | | | | | |
|----|--------|-------|------|------|-------|-------|------|------|--------|------|------|
| | INSAT-3D | 59.05 | 8.58 | 0.14 | -1.92 | -1.71 | 1.27 | 0.91 | 0.37 | 0.83 | 0.91 |
| | RS | 60.92 | 9.47 | 0.15 | -1.00 | | | | | | |
| PS | NOAA | 43.14 | 6.81 | 0.15 | 0.10 | | | | | | |
| | INSAT-3D | 55.68 | 2.36 | 0.04 | 0.05 | -2.5 | 1.55 | 0.77 | -0.002 | 0.45 | 0.92 |
| | RS | 55.66 | 3.44 | 0.06 | -1.00 | | | | | | |


## 4.2 Comparison of spatially distributed INSAT-3D TPW with Actual Rainfall Observation


Figure 6, shows the overall spatial distribution of rainfall change for different INSAT-3D TPW values during June to September 2016. This figure is constructed for all the daily rainfall amount from actual IMD observation between 0mm to 140 mm occurring over the stations and collocated mean INSAT-3D TPW values between 0mm to 90mm. It can be seen from the figure 6 that higher rainfall amount is accounted with higher INSAT-3D TPW values. However, atmospheric constituents and synoptic scale of monsoon conditions are an important factor when considering the occurrence of rainfall and satellite derived TPW. It is well demonstrated from the figure 6, that the heavy (60-80 mm) and heavy to very heavy rainfall (above 80mm) is very well matching with the higher TPW values. This can be related to the fact that the heavy rainfall occurs in the presence of higher TPW values (Wu et al, 2003). However, for the light to moderate rainfall amount (less than 40 mm) INSAT-3D TPW is not picking up the intensity, this may be attributed to the failure of capturing moisture divergence and subsidence in the lower levels of the atmosphere by the geostationary satellites due to its resolution. The ground resolution of INSAT-3D sounder is 10km.These limitations of sounders put the emphasis on high horizontal resolution (better than 10km), and on high vertical resolution (about 1km). This will provide frequent information on the 3D structure of atmospheric temperature and humidity, for the whole Earth disk seen by the satellite (except in and below clouds). They will be used, together with the imagers, to produce high resolution clouds or water vapour features, to track rapidly evolving phenomena. They are also designed to have an important role in the frequent observation of atmospheric chemical composition.






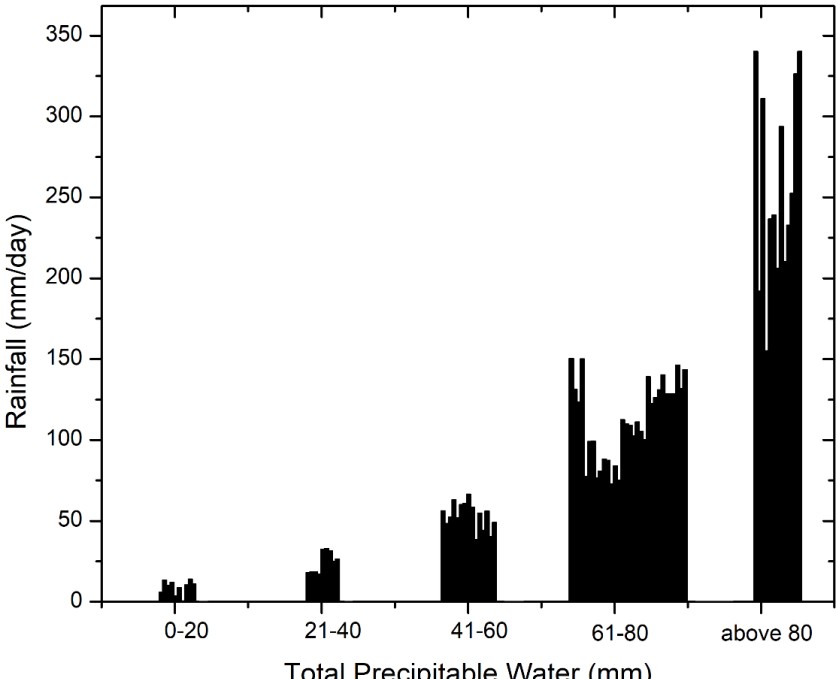

**Figure 6. Overall spatial distribution of rainfall with INSAT-3D TPW**




**4.3 A case study of INSAT-3D TPW with ground base GNSS TPW**
In this case study, hourly INSAT-3D sounder derived TPW, and GNSS TPW were analyzed for a
thunderstorm event occurred at Pune on 03.06.2017 of 1200UTC.The advantage of GNSS is
having access to multiple satellites, redundancy and availability at all times. Though satellite
systems don't often fail, if one fails GNSS receivers can pick up signals from other systems.
Figure 7, shows the hourly comparison between INSAT and GNSS TPW. The red bar shows the
time of occurrence (i.e., 1200 UTC) of thunderstorm over Pune city. It was observed from the
satellite imageries that initial convection development starts at 0600 UTC with multiple significant
convection seen over the area. It can be seen from the figure-7 that the INSAT-3D TPW is showing
the higher TPW values around 53 mm in comparison with GNSS TPW of 54 mm at 0600 UTC.
The higher TPW of INSAT-3D continues up to 1100 UTC with agreement with GNSS TPW. The
thunderstorm was reported at 1200 UTC. Since INSAT-3D retrieval cannot be made over cloudy





region, the TPW observation was not available after the 1200 UTC. However, prior to the event
INSAT-3D TPW can be considered as a precursor for mesoscale activity.

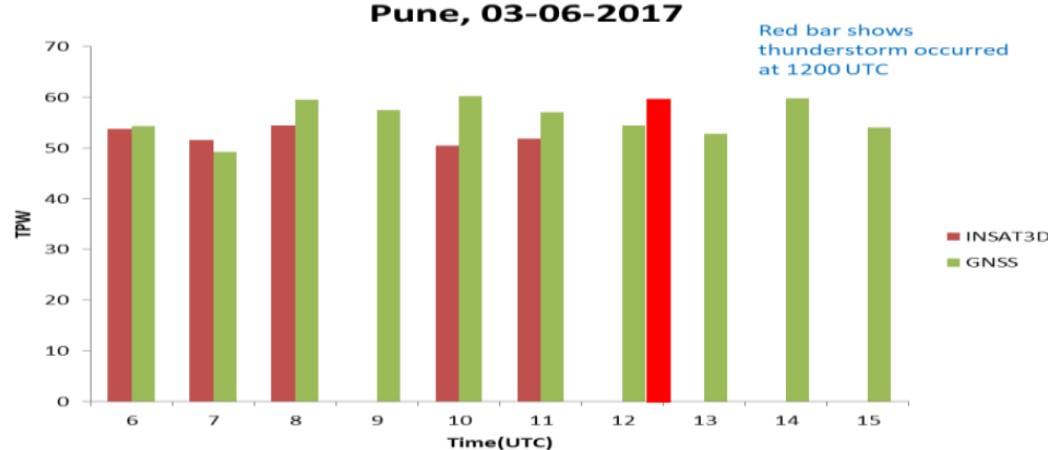


**Figure 7. A thunderstorm weather event on Pune (18.52°N 73.85°E) on 03.06.2017**

**CONCLUSION**
In the present study, INSAT-3D satellite sounder derived TPW and corresponding TPW from
radiosonde observations (RS) and National Oceanic and Atmospheric Administration (NOAA) N-
18 and N-19 have used to assess retrieval performances covering the entire season with
geographical region over the Indian sub-continent. The INSAT-3D derived TPW has also very
well picked the general feature of monsoon at daily as well as monthly scale. Most of the region
except north eastern parts and north western region the magnitude is slightly differ. This may be
attributed to the orography area of that region as well as geometry. It is also to be mentioned that
the INSAT-3D TPW on monthly scale show very good agreement with the sub divisional scale
rainfall observational pattern indicating good reliability to use the INSAT-3D TPW for
advancement of the monsoonal pattern. The advantages of INSAT-3D TPW are the availability of
the real-time data over the Indian region due to higher spatial and temporal resolution as compared
to polar orbiting satellites. The quality of the TPW from INSAT-3D wills substantial improves the
now-casting as well as the rainfall prediction by assimilating into the NWP model. Furthermore,



the spatial distribution of INSAT-3D TPW with actual rainfall observation is also been
investigated and found that heavy rainfall events occur in the presence of high TPW values.
The improvement observed in the current INSAT-3D sounder products specifically TPW is mainly
attributed to the GSICS bias corrections applied to the sounder radiances at IMDPS by SAC/ISRO,
Ahmadabad. This study will provide the reasonable interpretation and quantitative benefit to NWP
and can also offer substantial opportunities for improvement in now casting studies.

## ACKNOWLEDGMENTS

Authors are very much grateful and rendered by Director General of Meteorology Dr. K. J.
Ramesh, and given valuable suggestions. We specially thank C.M Kistawal and P. Thapliyal for
the improvement of INSAT-3D sounder retrievals specially the applying of the GSICS corrections
at IMDPS for the improvement sounder retrievals and their technical inputs. First authors also
thankful to NOAA for providing us required data and GSICS members for providing technical
support.

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

404   14,937.