# Peer review of "Potential of INSAT-3D Sounder Derived Total Precipitable Water"

_Atmospheric Measurement Techniques, 2018_

## Referee Comment (RC1) · Anonymous Referee #1 · 14 Jun 2018

The paper entitled 'Potential of INSAT-3D Sounder Derived Total Precipitable Water 1 Product for Weather Forecast' by Parihar et al. deals with the validation of INSAT-3D derived TPW dataset with other datasets at different temporal scales. With the limitations in deriving the TPW, the authors also showed its potential in predicting the thunderstorms by considering a case study. There is nothing new in the authors findings as INSAT-3d datasets are already validated against with different other datasets at different temporal scales. Also many existing articles are available that shows the importance of water vapor in predicting the storms. The authors neither show any improvements in the retrieval of INSAT-3D TPW nor show its applicability in statistical sense. Thus, I recommend 'Rejection' in its present form. Major concerns: 1) The motivation to the work is not clear as Ratnam et al. (2016) as already compared INSAT-3D

datasets with other datasets. 2) How come one time RS observation serve as a representative of daily mean? 3) INSAT-3D PWV measurements are available only during cloud free conditions then how the authors compared rainfall vs. pwv? 4) Literature survey is very poor 5) With one case study the authors are claimed that high TPW values can be used as a precursor to forecast thunderstorm. Is it true for all the thunderstorm cases as well as all high TPW will lead to thunderstorms? 6) The English is also very poor and difficult to follow.

———————————————————

---

## Short Comment (SC1) · 2 Jul 2018

The present program is basically shown the improvement notice after the GSICS corrections applied by Space Application Center, Indian Space Research Organization, Ahmedabad. In the abstract line 28 to 32 mentioned clearly that the corrections of GSICS have been applied at IMDPS, New Delhi for derivation of the products of INSAT-3D satellite and TPW is one such product. Therefore I am not satisfied with rejection in present form of the manuscript, in fact, this work should be encouraged for better improvement. However, INSAT-3D dataset was not initially GSICS corrected as in the case of current manuscript have shown. 1. IMD uses INSAT-3D sounder data particularly TPW, temperature and moisture profile issuance of nowcasting. Here no other dataset is readily available at the time of the forecast. In this condition, only INSAT-3D

sounder TPW with the higher temporal resolution of one hour is available to forecast-ers. The current TPW from INSAT-3D satellite can be utilized operationally for weather purpose and it can also offer substantial opportunities for improvement in now casting studies.

2. Basically for comparison of TPW and Rainfall by Wu. et. al., 2003. However, in our comparison, we excluded all the cloudy part. We considered only available TPW as suggested by Wu. et. al., 2003.

3. The recent literature along with their details and results in form of referees will be added into the modified version of the manuscript.

4. More number of case studies will be added into the modified version of the manuscript.

5. The manuscript will be rephrase and English will be improve in modified version.

Please also note the supplement to this comment:
https://www.atmos-meas-tech-discuss.net/amt-2018-8/amt-2018-8-SC1-supplement.pdf

**Supplement:**

The present program is basically shown the improvement notice after the GSICS corrections applied by Space Application Center, Indian Space Research Organization, Ahmedabad. In the abstract line 28 to 32 mentioned clearly that the corrections of GSICS have been applied at IMDPS, New Delhi for derivation of the products of INSAT-3D satellite and TPW is one such product. Therefore I am not satisfied with rejection in present form of manuscript, in fact this work should be encourage for better improvement.

However, INSAT-3D dataset was not initially GSICS corrected as in the case of current manuscript have shown.

1. IMD uses INSAT-3D sounder data particularly TPW, temperature and moisture profile issuance of nowcasting. Here no other dataset is readily available at the time of forecast. In this condition only INSAT-3D sounder TPW with higher temporal resolution of one hour is available to forecasters. The current TPW from INSAT-3D satellite can be utilized operationally for weather purpose and it can also offer substantial opportunities for improvement in now casting studies.

2. Basically for comparison of TPW and Rainfall by Wu. et. al., 2003. However, in our comparison we excluded all the cloudy part. We considered only available TPW as suggested by Wu. et. al., 2003.

3. The recent literature along with their details and results in form of referees will be added into the modified version of manuscript.

4. More number of case studies will be added into the modified version of manuscript.

5. The manuscript will be rephrase and English will be improve in modified version.

---

## Referee Comment (RC2) · S. Nath (Referee) · 8 Jul 2018

Paper entitled: 'Potential of INSAT-3D Sounder Derived Total Precipitable Water Product for Weather Forecast', this study showed validation of the INSAT-3D satellite derived product total precipitable water (TPW) dataset with radiosonde (RS), NOAA derived TPW, rain measured by rain gauges and one case study using Global Navigation Satellite System (GNSS). This work has done with different temporal scales and area with statistics. Study represents the capability of INSAT-3D sounder derived product and benefits for weather forecasting. Interesting to see that applying of GSICS correction to the sounder retrievals has impacted in the improvement of TPW products. INSAT-3D is geostationary satellite with first time sounder payload facility, keep in mind with this regard, this paper work is contiguous idea within the scope of Atmospheric

[Figure]

Measurement Technique Journal. I recommend for publication but the following points have to illustrate my concern: 1. Give full abbreviation of IMDPS in abstract and PB section 2.2. 2. In section 2.4, Is GISCS is providing any coefficients? Author should provide clear information about this. 3. In section 3., Has 50km square area been considered? 4. In section 4.1, comparison of INSAT-3D and RS at daily, monthly and subdivional scale then why is not promising over northern Indian region as comparison of southern region of India? 5. In section 4.2, Comparison of spatially distributed INSAT-3D TPW with Actual Rainfall observation, there should be more detail about the figure 6 that how it has constructed? 6. In section 4.3, A case study of INSAT-3D TPW with ground base GNSS TPW has been showed. For the justice of this research (prior to the eventINSAT-3D TPW can be considered as a precursor for mesoscale activity),author should give other case study too. It is strongly recommended that author should give one more case study of similar weather event.

---

## Author Comment (AC2) · 24 Jul 2018

**Response to the Referee #1**

**Referee's comment:** There is nothing new in the authors findings as INSAT-3d datasets are already validated against with different other datasets at different temporal scales. Also many existing articles are available that shows the importance of water vapor in predicting the storms. The authors neither show any improvements in the retrieval of INSAT-3D TPW nor show its applicability in statistical sense. Thus, I recommend 'Rejection' in its present form. Major concerns: 1) The motivation to the work is not clear as Ratnam et al. (2016) as already compared INSAT-3D datasets with other datasets.

**Authors' response:** It is to be noted that the India Meteorological Department (IMD) utilises the INSAT series of satellite data for day to day weather forecast on an operational bases. The timely availability of data is very important for issuing forecasting and nowcasting. To accomplish this on daily, monthly, and sub-divisional scale; satellite derived product is required for users, disaster management group and other services. Over of the period of time, for accuracy of satellite products and its authenticity, a proper calibration is required. In the present work, the GSICS calibration corrections (Global Space-based Inter-Calibration System) on Infra-Red (IR) sounder channels are incorporated at INSAT-3D Meteorological Data Processing System (IMDPS). TPW is derived using this corrected radiance. Subsequently, comparison of TPW with various other dataset is carried out for the validation purpose. This aims to produce corrections, ensuring the data consistency and allowing them to be used to produce globally homogeneous products for environmental monitoring.

In this paper, the analysis and validation justify the usefulness of current TPW product from INSAT-3D which was not exclusively studied by any other past study. Utilization of TPW product from INSAT-3D sounder is mainly in the nowcasting mode, operationally for weather purpose and it can also offer substantial opportunities for improvement in now casting studies. It is to be noted that TPW product utilised in the study incorporates the GSICS calibration corrections.

**Referee's comment:** How come one time RS observation serve as a representative of daily mean?

**Authors' response:** Twice observations over a day using RS were used for the comparison with daily TPW of INSAT-3D. Indeed, INSAT-3D TPW is not daily mean. Each RS was paired with

closest INSAT-3D TPW and patterned according to criteria suggested in Fuelberg and Olson (1991). The collocation criteria for INSAT-3D retrievals with RS and NOAA data are based on the following: (1) The absolute distance between the position (latitude and longitude) of the RS and the INSAT-3D retrievals has been considered as 0.5º (50 Km). This will minimize the differences arising from horizontal gradients. (2) The temporal difference between two sets of data is around ±120 minutes depending on retrievals and location of the RS station. (3) The INSAT-3D/RS were matched at 0000 and 1200 UTC (refer line no. 172 to 179).

**Referee's comment:** INSAT-3D PWV measurements are available only during cloud free conditions then how the authors compared rainfall vs. pwv?

**Authors' response:** We use the mean TPW of INSAT-3D sounder while comparing with the rainfall/rain rate. Rainfall accumulated over a given day is compared with mean TPW of that day, if sky found to be clear over that day. When water vapor reaches to its saturation level in the troposphere, it becomes conducive for occurrence of rain. The higher TPW is expected prior/around the event of rain and vice-versa. Thus, the positive association between TPW and rainfall is obvious. Yes, since only clear-sky TPW is under consideration, there won't be one-to-one correspondence with rainfall. It is the limitation of this comparison.

**Referee's comment:** Literature survey is very poor.

**Authors' response:** Literature review has been updated (refer line no from 51 to 54).

**Referee's comment:** With one case study the authors are claimed that high TPW values can be used as a precursor to forecast thunderstorm. Is it true for all the thunderstorm cases as well as all high TPW will lead to thunderstorms?

**Authors' response:** As suggested by the reviewers, two more case study of thunderstorms has been included (refer line no from 275 to 303) in the modified manuscript. It can be seen that most of the thunderstorms analysis have good signature prior to the occurrence of weather events. This can be mentioned here that, IMD (Forecasters, FDP Storm, http://nwp.imd.gov.in/fdp_now/) is

regularly utilizing these data in pre-monsoon season for nowcasting services over the Indian region.

It was a mistake to consider higher TPW as a precursor to forecast thunderstorm. But along with other meteorological parameters (e.g., CAPE), higher TPW observed during thunderstorm events can be utilized for studying such events.

**Referee's comment:** The English is also very poor and difficult to follow.

**Authors' response:** The English correction is made and authors think that the manuscript is improved significantly.

---

## Author Comment (AC3) · 24 Jul 2018

[revised manuscript text omitted]

14,937.

---

## Author Comment (AC5) · 29 Jul 2018

**Response to the Referee #2**

Paper entitled: 'Potential of INSAT-3D Sounder Derived Total Precipitable Water Product for Weather Forecast', this study showed validation of the INSAT-3D satellite derived product total precipitable water (TPW) dataset with radiosonde (RS), NOAA derived TPW, rain measured by rain gauges and one case study using Global Navigation Satellite System (GNSS). This work has done with different temporal scales and area with statistics. Study represents the capability of INSAT-3D sounder derived product and benefits for weather forecasting. Interesting to see that applying of GSICS correction to the sounder retrievals has impacted in the improvement of TPW products. INSAT-3D is geostationary satellite with first time sounder payload facility, keep in mind with this regard, this paper work is contiguous idea within the scope of Atmospheric Measurement Technique Journal. I recommend for publication but the following points have to illustrate my concern:

**Referee's comment:** Give full abbreviation of IMDPS in abstract and PB section 2.2.

**Authors' response:** This has been corrected from line no. 16 to 17 and 106 to 107.

**Referee's comment:** In section 2.4, Is GISCS is providing any coefficients? Author should provide clear information about this.

**Authors' response:** Yes. GSICS coefficients generated and corrections applied by Space Application Centre (ISRO), Ahmedabad. The corrections of GSICS coefficients are routinely applied at IMDPS, New Delhi for derivation of the products of INSAT-3D satellite and TPW is one of such product. Refer line no. from 140 to 148.

**Referee's comment:** In section 3., Has 50km square area been considered?

**Authors' response:** We have considered 50 km around the area from the Radiosonde Station place. In this methodology, each RS was paired with closest INSAT-3D retrievals and patterned according to criteria suggested in Fuelberg and Olson (1991). The collection criteria for INSAT-3D retrievals with RS data are based on absolute distance between the position (latitude and longitude) of the RS and the INSAT-3D retrievals has been considered as 0.5 (50 km). This will minimize the differences arising from horizontal gradients (Line no from 172 to 179).

**Referee's comment:** In section 4.1, comparison of INSAT-3D and RS at daily, monthly and subdivional scale then why is not promising over northern Indian region as comparison of southern region of India?

**Authors' response:** The comparison of INSAT-3D and RS over northern Indian region shows correlation coefficient of 0.87 which is comparable to that over southern region of India (i.e. 0.92). There is, indeed, very small difference between the observed correlation coefficient over these two regions. This difference could be attributed to number of points under consideration, averaging effect and uncertainty in the satellite retrieved TPW. (Refer line no 248 to 250, table 3)

**Referee's comment:** In section 4.2, Comparison of spatially distributed INSAT-3D TPW with Actual Rainfall observation, there should be more detail about the figure 6 that how it has constructed?

**Authors' response:** We use the mean TPW of INSAT-3D sounder while comparing with the rainfall/rain rate following the Wu et al 2003 . Rainfall accumulated over a given day is compared with mean TPW of that day, if sky found to be clear over that day. When water vapor reaches to its saturation level in the troposphere, it becomes conducive for occurrence of rain. The higher TPW is expected prior/around the event of rain and vice-versa. Thus, the positive association between TPW and rainfall is obvious. Yes, since only clear-sky TPW is under consideration, there won't be one-to-one correspondence with rainfall. It is the limitation of this comparison.

**Referee's comment:** In section 4.3, A case study of INSAT-3D TPW with ground base GNSS TPW has been showed. For the justice of this research (prior to the eventINSAT-3D TPW can be considered as a precursor for mesoscale activity), author should give other case study too. It is strongly recommended that author should give one more case study of similar weather event.

**Authors' response:** As suggested by the reviewers, two more case study of thunderstorms has been included (refer line no from 275 to 304) in the modified manuscript. It can be seen that most of the thunderstorms analysis have good signature prior to the occurrence of weather events. This can be mentioned here that, IMD (Forecasters, FDP Storm,http://nwp.imd.gov.in/fdp_now/) is regularly utilizing these data in pre-monsoon season for nowcasting services over the Indian region. However, It was evident that during monsoon season due to the straticumulus clouds over land region, the TPW sometime under/over estimating the actual rainfall. The orographic and

coastal region moisture (due to sea breezes) also not very well picked up by sounder derived TPW because of its coarser resolution. Therefore, along with other meteorological parameters (e.g., CAPE, CINE and other indices), higher TPW can be taken as one of the precursors during thunderstorm events can be utilized for studying such events.

---

## Author Comment (AC6) · 29 Jul 2018

[revised manuscript text omitted]

14,937.

---

## Author Response (AR2)

**Author's Response**

**Abstract:**

**Reviewer's Comments (RC):** Line 13: "..infrared radiances"

**Author's Response (AR):** This has written in line no. 13.

**Introduction:**

**RC:** Be consistent in using AMT reference style.

**AR:** AMT reference style has been incorporated in entire manuscript.

**RC:** Section 2. "Datasets"

**AR:** Replaced with "Datasets", refer line no. 80

**RC:** 2.1 Remove "in IMD"

**AR:** Removed "In IMD" refer line no. 81

**RC:** 2.4 Description of NOAA satellite data is too short and inadequate. Provide more information on the dataset, the retrieval procedure with appropriate references.

**AR:** The NOAA (National Oceanic and Atmospheric Administration) Satellite and Information Service provides timely access to global environmental data from satellites. In this study, we used blending TPW from two satellite sources, one from the Advanced Microwave Sounding Unit (AMSU) instruments on NOAA satellites (Ferraro et al., 2005), and the other from the Special Sensor Microwave Imager (SSM/I) instruments on Defence Meteorological Satellite Program (DMSP) satellites. In the blended Total Precipitable Water (TPW) product, individual biases of the data sources have been mitigated to produce a more meteorologically significant product. Blending retrieval procedure has detailed (Kidder et al., 2007) and methodology provides seamless global coverage without gaps to allow for the analysis of atmospheric moisture over land and ocean (Schmit et al., 2002 and Smith et al., 2007). The products are on a Mercator projection with 16 km resolution at the equator. The products are hourly in HDF-EOS file format. These operational products were produced by the NOAA/NESDIS (National Environmental Satellite, Data, and Information Service) Office of Satellite and Product Operations (OSPO). Refer line no 130-143.

**RC:** Figure 2. Areas marked with ellipses represent different sub-division scale. Mention it in the caption.

**AR:** Replaced by "Figure 2. Radiosonde Stations (red dots) of IMD over India. Areas marked with ellipses represent different sub-divisions." Refer line no. 110-111 and 118-119.

**Section 3. Methodology:**

**RC:** Line 151-153: Provide appropriate references.

**AR:** It has been mentioned. Refer line no. 163-165.

**RC:** g is the gravity constant. Consider sentence restructuring

**AR:** The sentence has restructured, refer line no 182.

**Section 4. Result and Discussion:**

**RC:** Section 4.1 "…Sub Division Scales"

**AR:** Refer line no. 194-195

**RC:** Figure 3 caption: "Comparison of INSAT-3D derived TPW with that from RS network.."

**AR:** Replaced by "Figure 3. Comparison of INSAT-3D derived TPW with RS observed TPW from May to September 2016". Refer line no. 204-205

**RC:** Expand the discussion little more. Describe the overall uncertainty in the TPW retrievals and associated sources of errors.

**AR:** INSAT-3D TPW is able to measure the synoptic features of weather phenomena at monthly scale over the Indian region very well. However magnitude differs, it can be termed as source of error due to registration and navigation issues during the night time. The consistent and better correlation has seen above 40 mm of TPW values, whereas for less than 40 mm TPW values, INSAT-3D underestimates slightly. This is due attributed to seasonal variation, orographic of the region and different climatic zone over India. Refer line no 207-213.

The mean difference between RS and INSAT-3D TPW is much higher in the month of July 5.57 mm. It is due to the substantial rainfall during the monsoon season and in the subsequent months August and September 5.24, 5.3 mm respectively. It was also reported by Ratnam et al., 2016 that mean difference in the water vapor is as high as 20-30%. The dry basis of 10-25% in INSAT-3D channel compare to similar satellite and reanalysis dataset was also noted. Refer line no 224-229.

**RC:** Figure 4. Instead of denoting each sub-plot with a), b), c)…denote it with the actual month of observations; it increases the readability of the plot.

**AR:** Replaced "Figure 4. INSAT-3D sounder TPW with RS for May, June, July, August and September 2016" Refer line no 242-243.

**RC:** INSAT-3D TPW is consistently lower for values < 40 mm for which some discussion is needed.

**AR:** INSAT-3D TPW is able to measure the synoptic features of weather phenomena at monthly scale over the Indian region very well. However magnitude differs, it can be termed as source of error due to registration and navigation issues during the night time. The consistent and better correlation has seen above 40 mm of TPW values, whereas for less than 40 mm TPW values, INSAT-3D underestimates slightly. This is due attributed to seasonal variation, orographic of the region and different climatic zone over India. The largest differences are observed mainly over mountainous areas and/or near the sea, which reveal differences in representativeness. Good confidence in INSAT-3D TPW estimates is gained during periods of moderate to heavy rain. Refer no 207-215.

**RC:** Figure 5. Improve the caption

**AR:** Written "Figure 5. Comparison of INSAT-3D derived with RS and NOAA observed TPW sub-division scales NI, WI, CI, WI& PS from May to September 2016". Refer line no 267-268.

**RC:** Section 4.2 "Comparison of INSAT-3D TPW with Actual Rainfall Observations"

**AR:** Replaced by "Comparison of INSAT-3D TPW with Actual Rainfall Observation". Refer line no. 272

**RC:** Figure 6. Showing the data as "box and whisker" plot would enhance the readability of the plot. The data arranged in Figure 6 is somehow unclear. What is the area coverage considered for such comparison? Over the entire Indian sub-continent? It must be stated in the text as well as in the figure caption.

**AR:** Replaced Figure 6 caption "Figure 6. The Box-Whisker plot for comparison of INSAT-3D TPW with actual rainfall over Indian region" Refer line no 299-300.

The box-whisker plot shown in the Figure 6 compares the actual rainfall observation and INSAT-3D TPW for different values during June to September 2016. This figure is constructed from the daily rainfall observation between 0 to 140 mm occurring over the 34 stations and collocated mean INSAT-3D TPW values between 0 to 90 mm over the entire Indian region. It can be seen from the Figure 6, that TPW is binned for the ranges 0-20, 21-40, 41-60, 61-81 and >80. As seen from the whiskers, the rainfall has least scatter for the 0-20 bin, while for TPW >80 it shows most scatter. The mean and median are almost same for all the TPW bins, except for the TPW >80. There exist exponential behavior between rainfall amounts with higher INSAT-3D

TPW values. However, further analysis with more number of observations is required for the quantification of non-linear/exponential relationship. Refer line no 273-282.

**RC:** Line 265: Remove "during summer monsoon season"

**AR:** It has removed.

**RC:** The rainfall-TPW relationship appears to exhibit an exponential behavior, which author completely misses to mention in the discussion. Does any other paper explore such relation or already noted such behavior?

**AR:** The mean and median are almost same for all the TPW bins, except for the TPW >80. There exist exponential behavior between rainfall amounts with higher INSAT-3D TPW values. However, further analysis with more number of observations is required for the quantification of non-linear/exponential relationship. Refer line no 276-282.

**RC:** Also, author should mention here that TPW corresponds to the cloud-free observations and rainfall measurements are for cloudy atmosphere.

**AR:** The TPW corresponds to the cloud-free observations and rainfall measurements are for cloudy atmosphere. Refer line no 285-286.

**RC:** Section 4.3, Provide latitude & longitude in parenthesis

**AR:** Pune, Kochi and Dibrugarh respectively latitude and longitude 18.52°E 73.85°N on June 3, 2017 at1200UTC, 9.93°E 76.26°N on June 6, 2017 at 0600UTC and 27.47°E 94.91°N on June 9, 2017 at 0000UTC. Refer line no. 304-306.

**RC:** Line 282: Can author provide the URL link to the satellite images referred here?

**AR:** Further details can be found at http://gnss.imd.gov.in/TrimblePivotWeb/ . Refer line no 307-308

**RC:** Line 284: "Figure 7" (not figure-7)

**AR:** Replaced by "Figure 7". Refer line no 313.

**RC:** Line 299: "…compares reasonably well with the GNSS TPW observations"

**AR:** Refer line no 331-332.

**RC:** Indicate dates in this format throughout the manuscript: Jan 1, 2018

**AR:** Yes. Date formats are incorporated in whole manuscript. Refer line no 330.

**RC:** Please explain the sudden dive in TPW from ~50 mm to 20~ over Dibrugarh?

**AR:** At 0000 UTC of thunderstorm over Dibrugarh city was reported. The initial convection development started at 1800 UTC with values around 53 mm in comparison with GNSS TPW of

58 mm at 1800 UTC. It can be very well seen from Figure 7, dive has noted at 1400 UTC with values 24 mm from 50 mm. This is due to less precipitation occurred in Dibrugarh while 1400 to 1800 UTC, no precipitation has noted due to cloudy sky. Refer line no. 324-328.

**Conclusion:**

**RC:** Line 311-314: Rephrase the sentence something like "In the present study, we have assessed the retrieval performance of INSAT 3D TPW by comparing it with corresponding observations from radiosonde network and NOAA' GNSS network over the Indian region".

**AC:** Sentence has incorporated, refer line no 343-345.

**RC:** Line 320: "…TPW product in forecasting advancement of monsoon precipitation over the Indian region"

**AR:** Refer line no. 354-355.

**RC:** Line 322-324: "INSAT 3D TPW product offers near-real time availability over the Indian region with higher spatial and temporal resolution compared to the other derived from polar orbiting satellites"

**AR:** INSAT 3D TPW product offers near-real time availability over the Indian region with higher spatial (resolution 10 km) and temporal resolution (60 min) compared to the other derived from polar orbiting satellites. Refer line no. 358-359.

**RC:** Mention here the spatial and temporal resolution of INSAT 3D TPW product.

**AR:** Refer line no 358-359.

**RC:** The write up for Conclusion seems short. Author needs to discuss the comparison results in little more details.

**AR:** Refer line no. 348-349 and 361-363.

**Acknowledgments:**

**RC:** Authors are grateful to Dr. K. J. Ramesh, the Director General of Meteorology IMD for offering valuable suggestions.

**AR:** Refer line no 365-366.

**RC:** Rephrase the sentence as "We appreciate the work of C. M. Kistawal and P. Thapliyal of applying GSCIS correction at IMDPS for improving sounder retrievals. We thank both them for providing their technical inputs."

**AR:** Refer line no 366-368.

**RC:** The first author also thanks NOAA for providing satellite data of TPW used in the comparison against that of INSAT 3D sounder.

**AR:** Refer line no 368-369.